# Mortar Bond Strength: A Brief Literature Review, Tests for Analysis, New Research Needs and Initial Experiments

**DOI:** 10.3390/ma15062332

**Published:** 2022-03-21

**Authors:** Janaina Salustio, Sandro M. Torres, Anne C. Melo, Ângelo J. Costa e Silva, António C. Azevedo, Jennef C. Tavares, Matheus S. Leal, João M. P. Q. Delgado

**Affiliations:** 1Engineering Department, Federal Rural University of the Semi-Arid (UFERSA), Angicos 59515-000, Brazil; janaina.salustio@ufersa.edu.br; 2Technology Center, Postgraduate Program in Civil and Environmental Engineering, Federal University of Paraíba (UFPB), João Pessoa 58051-900, Brazil; jennef.carlos@ufersa.edu.br; 3Departament of Mechanical Engineering, Federal University of Paraíba (UFPB), João Pessoa 58051-900, Brazil; sandromardentorres@yahoo.co.uk; 4Postgraduate Program in Material Science and Engineering, Federal University of Paraíba (UFPB), João Pessoa 58051-900, Brazil; anne.carolinemelo@hotmail.com; 5Department of Civil Engineering, Catholic University of Pernambuco, Recife 50050-900, Brazil; angelo.silva@unicap.br; 6Recife Campus, Federal Institute of Education Sciences and Technology of Pernambuco (IFPE), Recife 50740-545, Brazil; antonio.costaazevedo@fe.up.pt; 7CONSTRUCT-LFC, Departamento de Engenharia Civil, University of Porto, Rua Dr. Roberto Frias, 4200-465 Porto, Portugal; 8Department of Science and Technology, Federal Rural University of the Semi-Arid (UFERSA), Carnaúbas 59780-000, Brazil; 9Departament of Civil Engineering, Federal University of de Campina Grande, Campina Grande 59780-000, Brazil; matheus3993@gmail.com

**Keywords:** adhesive, slots, bond strength, cyclic stresses, elastic energy, experimental campaign

## Abstract

Despite technological advances in the production processes of the materials for ceramic façade coatings, the problems of detachments are still frequent. Therefore, this work aims to investigate, through a literature review, the existing gaps related to the adhesion ability of adhesive mortars, identifying new research needs that can better explain the behavior of the material. In addition, an experimental procedure was developed to evaluate the mechanical capacity of adhesive mortars when subjected to cyclic stresses. Dynamic stress measurements are presented for several blocks of mortar and on similar blocks but with a slot drilled prior to measurements (intended to represent failure). From these data we calculated values of stress energy, elastic energy, and dissipated energy. The experimental results showed that the energy involved in the test process accompanied the load values and current stress values. The mortar samples with the previous failure absorbed and dissipated less energy than mortars without failure, showing that materials that have less energy to dissipate, are materials that have developed less capacity to adhere, that is, to keep their parts together.

## 1. Introduction

The lack or loss of adhesion of the ceramic tiles on buildings leads to economic losses, but also damages against the maintenance of the integrity and useful life of buildings, affecting the habitability, thermo-acoustic comfort, the protection of structures, and seals against the action of bad weather and their water tightness. In Brazil, the application of ceramic tiles on façades of multi-story buildings is widespread. According to the National Association of Ceramic Manufacturers [1], it is estimated that in 2020 Brazil produced around 840 million square meters of ceramic tiles, occupying the third position in world production (the world total global production volume of ceramic tiles amounted to 16 thousand million square meters, in 2020). Ahead of Brazil are China, which produced more than 5 thousand million square meters of ceramic tiles, and India, which produced just over 1.2 thousand million square meters. Of this total, 179 million are used on walls and 18 million on façades in the country. Despite the wide application of ceramic coverings on facades and the technological evolution that provides the production of high-quality pieces and compliance with international standards, the tiles detachment is still a problem that persists. However, it should be noted that there are several factors related to the tile detachment problem, such as quality of materials, construction techniques used, substrate conditions, and environmental conditions.

In a survey carried out by Mansur et al. [2] on pathological manifestations in ceramic tile coverings, it was observed that in 84% of the buildings analyzed there was detachment with adhesive rupture at the ceramic tile/adhesive mortar interface. The explanation for the frequency of occurrences of displacement at this interface is that this region of the coating system is the most requested by shear forces when considering the thermal and hygroscopic expansion effects of the ceramic tiles.

In this sense, this work aims to study the mechanical adhesion capacity of adhesive mortars. The experimental program presented below is part of a study center on the efficiency of adhesive adhesion for ceramic façades and bonding at their interfaces. The results of this research consist of initial works on the subject, to know the behavior of type AC III adhesive mortar, through the action of cyclic loads that simulate the fatigue caused by the expansion and contraction movements of the adhesive when exposed to thermal variations, based on the evaluation of the energy involved in the process.

### 1.1. Adhesion Matrix Substrate

Several theories try to explain the adhesion formation mechanism. Costa [3] summarizes these theories in:−The theory of mechanical interlocking, which occurs when the adhesive material penetrates the imperfections on the solid surface (pores and roughness);−Chemical bond theory, when the level of adhesion is defined by the primary and secondary chemical bonds formed at the interface and whose bond strength depends on the contact between the phases and, the greater the contact, the greater the bond strength;−Transition zone theory, which explains the phenomenon of adhesion through the Formation of an interface in the region of adhesive-substrate contact;−Thermodynamic or adsorption theory which proposes that the adhesive adheres to the substrate due to the interatomic and intermolecular forces established at the interface after “wetting” the substrate by the adhesive.

This last theory also encompasses the concepts of rheology, wetting and surface energy; in addition to electrostatic and diffusion theories used for material systems that are beyond the scope of this work.

Petit et al. [4] showed that it is very difficult to discern the most important contribution of each adhesion mechanism, as the structure and properties of the substrate and the adhesive often contribute to the bonding of the adhesive to the substrate. The authors presented the optimum concentration of constituent components, cement, and organics that can be found to maximize mechanical property performance of cementicious tile adhesives.

In summary, the works presented showed that the explanation of the adhesion mechanism between two materials in contact is complex and involves many theories and many phenomena so the elaboration of a single theory is difficult. Thus, this article will discuss some concepts related to the theme of adhesion in order to analyze the parameters that influence the cementitious substrate-matrix contact.

### 1.2. Types of Adhesion

Starting with chemical adhesion, it is understood as being the result of molecular attraction forces (primary and secondary) between the phases. Secondary forces, such as van der Waals, formed mainly between molecules of material, are mainly responsible for the adhesion of multiphase materials, although they have lower binding energy than primary bonds. Even so, these bonds are strong enough to explain the adhesion, not being necessary to the occurrence of primary bonds between the two materials guarantee their stability [3].

Studies carried out by [4,5,6] proved that the use of re-dispersible powder-type polymers in cementitious materials improved their performance on the adhesion resistance of mortars. The strength gain was especially attributed to chemical interactions between cement and polymer compounds, the formation of the polymeric film, as well as the formation of hydroxyl during the hydrolysis process in the polymer.

Although there is a scientific basis demonstrating the chemical nature of adhesion, the contribution of mechanical interlocking cannot be ruled out, as it is quite widespread in the area of mortars as the main adhesion mechanism.

Thus, several studies over the years were carried out to prove the theory of mechanical interlocking. Paes et al. [7] concluded that mortars produced with aggregates of larger particles facilitate the transport of water from the mortar to the substrate. Papaoannou et al. [8], studying the adhesion between the settlement mortar and ceramic bricks for masonry, concluded that the blocks with the best adhesion results were those in which the presence of complex Ca-Al-Si phases, capable of penetrating the pores of the brick.

Botas et al. [9] performed microscopic analyzes at the interface of their aerial lime mortars for repairing historic buildings. The authors conclude their research by stating that there is a close relationship between the Ca/Si ratio and the values of resistance to adhesion found.

In cementitious substrate-matrix contact, there are also several studies, such as the works developed by [10,11,12,13,14,15], which aimed to understand and improve the adhesive’s ability to extend over the substrate. However, should be noted that there is still a need for studies aimed at improving the current model for assessing adhesion strength. On the other hand, that aim to seek parameters that had better explain the behavior of the materials that make up the coating system. This is because, despite all the progress already made, problems with debonding persist, even when the component materials of the coating meet current regulatory criteria.

### 1.3. Destructive Testing Methods

Among the destructive tests, the tensile bond strength test is the most widespread, with national (NBR 13528, and NBR 14081-4 [16,17]) and international standards (EN 1015, EN 12004-2, ASTM D 7234, and ASTM D 4541 [18,19,20,21]) which standardize the assay. However, Costa et al. [22] mention that the tensile adhesion test presents highly variable results due to factors inherent to the procedure itself, such as the angle and application of the load, the application speed, and the type of equipment.

However, studies carried out by [23,24,25,26,27], report that there is a lack of compatibility between the results of adhesion tests obtained in the laboratory and those obtained in the field. The authors demonstrated that the laboratory environment is a relevant factor in the higher average of adhesion, a fact attributed to greater control over the application of coatings and the performance of tests.

Costa et al. [22] and Lopes et al. [28] studied some factors that may influence the variability of the results of the tensile adhesion tests, as great variability in their results is observed in the literature, with coefficients of variation in the order of 10% to 35% being reached.

Another point to be considered when analysing the tensile bond strength test is the fact that it only takes into account the traction effort instead of the shear. According to Łukasik et al. [29], the evaluation through shear strength would better portray the forces to which the adhesive mortars are subjected, better classifying their performance in service.

For Silva [30], shear stresses occur especially due to thermal movements of the coating, which deform the layers that make it up due to their different expansion coefficients. The emergence of traction between layers of different materials also gives rise to shear stresses between the back of the pieces and the adhesive mortar, resulting in warping of the ceramic tile or swollen due to the gradual loss of strength.

Recently, a work in the area of mortar coatings, carried out by [15], evaluated the bond strength from the crack propagation test in stress mixed mode (MMF test), which simulates tensile and shear forces, simultaneously. The authors concluded that the results of tensile bond strength were higher than the results obtained by the MMF test, indicating an overestimation.

In accordance with the described above, it is possible to observe that the use of the tensile adhesion test, to assess the adhesion of coatings, needs to be complemented with other tests that assess the other efforts to which the coating layers are subjected. This is because this test does not take into account cyclical efforts, due to the variations in temperature and humidity to which the facades of buildings are exposed. Such variations can lead to material failure due to fatigue, requiring those other measurement methodologies should be used for a more realistic assessment of the effectiveness of sticky materials.

In this sense, there is a need for research in the area of coatings to be developed, to fill the gaps in the effective analysis of the adhesion capacity of these layers, considering the signs of detachment on the facades of buildings are still recurrent. Therefore, it is necessary to understand and study the variation in the flexibility of the materials that are used in the facade during its crack propagation process, of exposure to fatigue, using established techniques that more realistically simulate the efforts to which coating layers are subjected and analyze parameters based on energetic principles. Research conducted by Champaney et al. [31,32,33] studied damage mechanisms and formulated models of loss of adhesion of adhesive materials from the analysis of critical fracture energy.

## 2. Materials and Methods

This work aims to analyze the mechanical adhesion capacity of the adhesive mortar. Several experimental tests were carried out in order to simulate the action of simultaneous tensile and shear stresses. The pre-existence of failures arising from the shrinkage of the material was also considered.

### 2.1. Materials

The experiments were carried out with prismatic test bodies of adhesive mortar with dimensions of 40 × 40 × 160 mm^3^. For this purpose, 42 samples were molded: 21 samples (Slot-B, Slot Blocks) were prepared by introducing a slot in the central axis of the sample (with 9 mm height, which corresponds to 22.5% of the specimen height), as shown in Figure 1A; and the other 21 samples (Solid-B, Solid Blocks) were prepared without the slot. The mortar samples were prepared according to the manufacturer’s guidelines. Finally, the mortar prisms were deformed 24 h after molding, and the prims were placed in moisture-curing for 28 days, in order to reach the age for performing the mechanical tests.

A slot drilled prior to measurements was used to induce a controlled slot providing a concentration of stresses and significantly reducing the formation of multiple crack points. The pre-existing slot also simulates defects imposed by the reduction in bonded areas in the ceramic wall cladding. These defects can originate either from the lack of complete filling of the ceramic tile by the adhesive, from slots produced by mechanical efforts, and dimensional thermal and/or hygroscopic variations.

This adequacy of the tensile test was based on the method for calculating the fracture energy of concrete proposed by RILEM 50 [34]. This method uses the three-point bending test of a beam with a notch (slot) installed in the middle of the central span; and the fracture energy (Gc) is defined as the energy required for the formation of a slot with a unit area, defined by the projection of a plane parallel to the main direction of the induced slot.

### 2.2. Methods

The industrialized adhesive mortar used for molding the 42 samples has a classification of ACIII (high resistance), according to NBR 14081-4 [17]. Table 1 presents the results of the test performed in order to characterize the adhesive mortar used.

In order to measure the ability to resist permanent deformation associated with the introduction of a slot in a sample, the dynamic modulus of elasticity modulus was determined using ultrasonic wave propagation in all manufactured samples. The experimental results obtained for Solid-B and Slot-B samples were 9.9 ± 0.7 GPa and 9.4 ± 0.6 GPa, respectively. These results showed that the introduction of a previous slot in the prismatic samples of adhesive mortar caused a reduction in its elastic modulus by approx. 6% and the difference between the mean of the two sample groups is statistically significant (*p*-value = 0.03), from the performance of the Student’s *t*-test, with a 95% confidence level.

In summary, to assess the mechanical performance of the adhesive mortar and the suitability of the test with cyclic loads, the following mechanical tests were performed:(a)Dynamic elasticity modulus, in accordance with the Brazilian standard NBR 15630 [35];(b)Tensile strength in static bending, according to the Brazilian standard NBR 13279 [36];(c)Tensile strength in dynamic bending, using cyclic loads.

In the literature, there are different methods to determine the dynamic modulus of elasticity (*E_d_*), such as the penetration resistance method; the maturity method; the ultrasonic pulse velocity method (UPV); the wave propagation method; the voltage for detection of voids, imperfections, and discontinuities in mass concrete; the resonance frequency method; the magnetic and electrical methods; the nuclear and reactive methods; the method of cyclic loading application and tomography of reinforced concrete [37,38].

In this work, the dynamic modulus of elasticity of the mortars was obtained through the propagation of longitudinal waves, obtained by ultrasonic pulses. The relationship between pulse velocity and dynamic modulus of elasticity is demonstrated through Equation (1):(1)$$E_{d}=\rho \cdot V^{2}\frac{\left(1+\nu \right)\cdot \left(1-2\nu \right)}{1-\nu }$$
where, *E_d_* is the dynamic modulus of elasticity of the mortar (N/m^2^), *ρ* is the mass density of the mortar (kg/m^3^); *V* is the wave propagation speed (km/s), and *ν* is Poisson’s ratio.

The mechanical tests with static loads for intact samples (Solid-B) were carried out according to NBR 13279 [36], in order to determine the tensile strength in three-point bending. For the Slot-B samples, an adaptation of the NBR 13279 [36] test was performed, incorporating the recommendations of the bending test proposed by [34]. The equipment used was the Shimadzu’s Autograph AGS-X Series Precision (Kyoto, Japan), as illustrated in Figure 1B.

To carry out the test with the application of cyclic loads, it was first necessary to establish the loading interval to be adopted. The adopted limits corresponded to 10% and 50% of the maximum load supported by the mortar groups during the test with a static load, respectively, for the lower and upper limits of the range. The resistance test with cyclic loads was carried out with the Servopulser Dynamic Testing machine (Shimadzu, Kyoto, Japan) from Shimadzu E-type Load Frame with the Software Controller 4890, and the samples were submitted to a load alternating frequency of 2 Hz. The bending tensile strengths of Solid-B and Slot-B and their elastic and dissipated energies were obtained, and the vertical displacements were registered by the test machine’s cursor. It should be emphasized that elastic energy is an important parameter to be evaluated, as it can be related to the mechanical behavior of the material under analysis.

The elastic energy was obtained from the area under the hysteresis curve and the data from the central section of the curve were considered to obtain the energy values. A line connecting the start and endpoints of the central section of the curve was used to perform the calculations. The dissipated energy was obtained by the difference of the elastic energies of the loading and unloading curves of the considered section. It used the central section of the curve due to the curve approximation (force-displacement) to a straight line, allowing the linearity associated with the characteristic of elasticity, and a better adjustment of the elastic energy values obtained.

The elastic energy (*E*_δ_) was calculated based on the concepts present in [39] and represented by Equation (2):(2)$$E_{\delta }=\int_{a}^{b}F^{*}\left(u\right)du\;$$
where $E_{\delta }$ is the elastic energy, obtained by calculating the area under the central stretch of the hysteresis curve (linear stretch), represented by the limits *a* and *b*. The curve was generated from the force (*F*) and displacement (*u*) data obtained experimentally. For each *u_i_* ∈ (*a*,*b*), the *F** was determined by Equation (3):(3)$$F*=\frac{F_{1}\left(u_{i}\right)+F_{2}\left(u_{i}\right)}{2}\;$$
where $F_{1}\left(u_{i}\right)\;\mathrm{and}\;F_{2}\left(u_{i}\right)$ are the values of the applied loads that generated the loading and unloading curves, respectively.

The dissipated energy, $E_{\gamma }$, also based on the concepts presented by [39], is represented, analytically by Equation (4):(4)$$E_{\gamma }=\int_{a}^{b}(F_{1}\left(u\right)-F_{2}\left(u\right))\;du\;$$
where $F_{1}\left(u\right)\;\mathrm{and}\;F_{2}\left(u\right)$, are, respectively, the loading and unloading curves on the hysteresis figure.

## 3. Results and Discussion

### 3.1. Bending Tensile Strength: Static Loading versus Cyclic Loading

The results of the load tests with static loading and cyclic loading are presented in Table 2 and Figure 2. The experimental results of the tests with static loading showed that the highest values of maximum load are verified in the samples without the presence of the fault (Solid-B). Such behavior shows that the presence of discontinuities in the material, such as voids, slots formed by retraction or from thermal and hygroscopic movements, or even failure to fill the ceramic tile arising from the application, impair the mechanical performance of the adhesive, which consequently will harm its bonding efficiency on ceramic tiles.

The presence of the previous slot reduced the adhesive’s load-bearing capacity by 60%. These results are in accordance with the research carried out by [15], who observed that the presence of adhesion extension failures in samples of adhesive mortars with ceramic tiles caused a reduction in the adhesion capacity of approximately 51.4%. The analysis of the mortar performance in terms of maximum-acting stresses showed that the reduction caused by the presence of the previous failure is smaller, in the order of 32%.

Figure 2 presents the hysteresis curves obtained from the experimental test with cyclic loads after 6000 cycles. Only six samples of Slot-B and three samples of Solid-B reached the limit of 6000 cycles, the other ones ruptured before reaching it.

In order to understand what would happen to samples when subjected to cyclic loads for as many cycles as possible, maximum load values were adopted, corresponding to 50% of the maximum load value reached in the static load test. The maximum values adopted for the Solid-B and Slot-B were, respectively, 896.8 N and 363.2 N.

As can be seen in the first cycles of Figure 2, the loading and unloading curves follow a very close trajectory, with their total deformation almost entirely recovered in the form of elastic deformation. However, instead of the first cycles, it is possible to observe that this elastic recovery starts to present a small reduction, with the continuity of the experimental tests. Even with load values significantly below the maximum value supported by the material, its internal discontinuities or fatigue promote stress-concentrating points that lead to the formation of micro-cracks and their subsequent nucleation and propagation, causing a failure in the material [40]. According to Rosa [41], materials even when tested below their elastic limit can undergo, when requested cyclically, permanent changes in their crystalline structure, with small plastic deformations, even if imperceptibly.

Related to the samples tested, 75% of Solid-B and 66.7% of Slot-B withstood, without breaking, until the end of the assay (6000 cycles). This fact confirms what was also observed in the static test. Table 3 presents the performance of the adhesive in terms of tension and energy, with data obtained from the central section of the curve (see Figure 3), for the Solid-B and Slot-B mortar samples, respectively.

The results presented in Table 3 showed that the values of stress and energy developed during the test remained practically constant throughout the cycles, acting within the previously established range. The values presented in the tables also showed that the energy involved in the material’s microstructural alteration process followed the values of the acting loads and tensions, which allows an analysis of the material’s mechanical performance to be carried out based on the analysis of the material’s energy performance. This type of analysis is preferable because it allows a global rather than a local evaluation of a material with a heterogeneous internal structure and presence of discontinuity, as in the case of the mortar sample.

The mortar samples with a pre-existing slot absorbed and dissipated less energy than the non-failed mortars. According to Grazzini et al. [42], the energy released by the material can be related to its loss of adhesion; in this way, materials that have less energy to dissipate are materials that have developed less ability to adhere, that is, to keep their parts together.

This statement corroborates what was already observed in the static load test, in which the presence of the flaw in the adhesive impaired its ability to distribute stresses along its cross-section, reducing its mechanical strength. The material with more energy to dissipate manages to reduce the propagation speed of the micro-cracks formed, maintaining the cohesion of its parts and reducing the concentration of stress. The argument presented justifies why the mortars with the previous failure had a smaller number of samples, reaching the 6000 cycles proposed in the test.

### 3.2. Influence of the Previous Slot

The introduction of previous slots in the mortar samples induced the propagation of a single slot, promoting a concentration of stresses and significantly reducing the formation of several crack points, as presented in Figure 4.

The presence of the previous slot allowed to simulate the presence of gluing failures or damage that appear due to the requests imposed on the facades. These failures compromise the effective area of contact of the adhesive with the ceramic tile, affecting the elastic capacity of the adhesive. For the samples tested an analytical study was performed and presented in Table 4, where I_c_ is the moment of inertia of the cross-section area of the beam, Mf_c_ and M_fd_ are the maximum moments for the loading and unloading cycles, respectively, and σ_c_ and σ_d_ are the normal stresses, operating in the region of maximum moments, for loading and unloading cycles, respectively.

In Table 4, the analytical analysis performed shows that, among the elastic properties analyzed for the bending test, the moment of inertia and the bending moments are the ones that suffered the greatest reductions due to the introduction of the slot in the specimens, favoring the reduction in the energetic performance of the material.

## 4. Conclusions

This work presents the results of an experimental campaign in order to understand the mechanical fastening capacity of ACIII type adhesive mortars. The mortar samples without and with previous failures were submitted to tests under static loads. Based on the experimental results obtained the main conclusions are:−The results with static load showed that the presence of the failure reduced by 60% the load support capacity of the adhesive mortar;−The adoption of the previous failure for material evaluation helped to simulate the presence of failures that compromise the effective area of contact of the mortar with ceramics, bringing a more realistic measure of the adhesive bonding capacity in the presence of previous defects;−The adaptation of the bending tensile strength test with the use of cyclic loading confirmed the results already obtained with static load, however, improving the determination of the load and stress values that the material can actually work without failure. The changes in the maximum values supported by the material without the material reaching the fault were evidenced by the need to adapt the loads adopted in the test;−The mechanical performance of the material was verified from an energetic point of view, and it was observed that the energy involved in the test process accompanied the values of the loads and current stresses;−Mortar samples with the previous failure absorbed and dissipated less energy than mortars without failure, showing that materials that have less energy to dissipate, are materials that have developed less capacity to adhere, that is, to keep their parts together.

## Figures and Tables

**Figure 1 materials-15-02332-f001:**
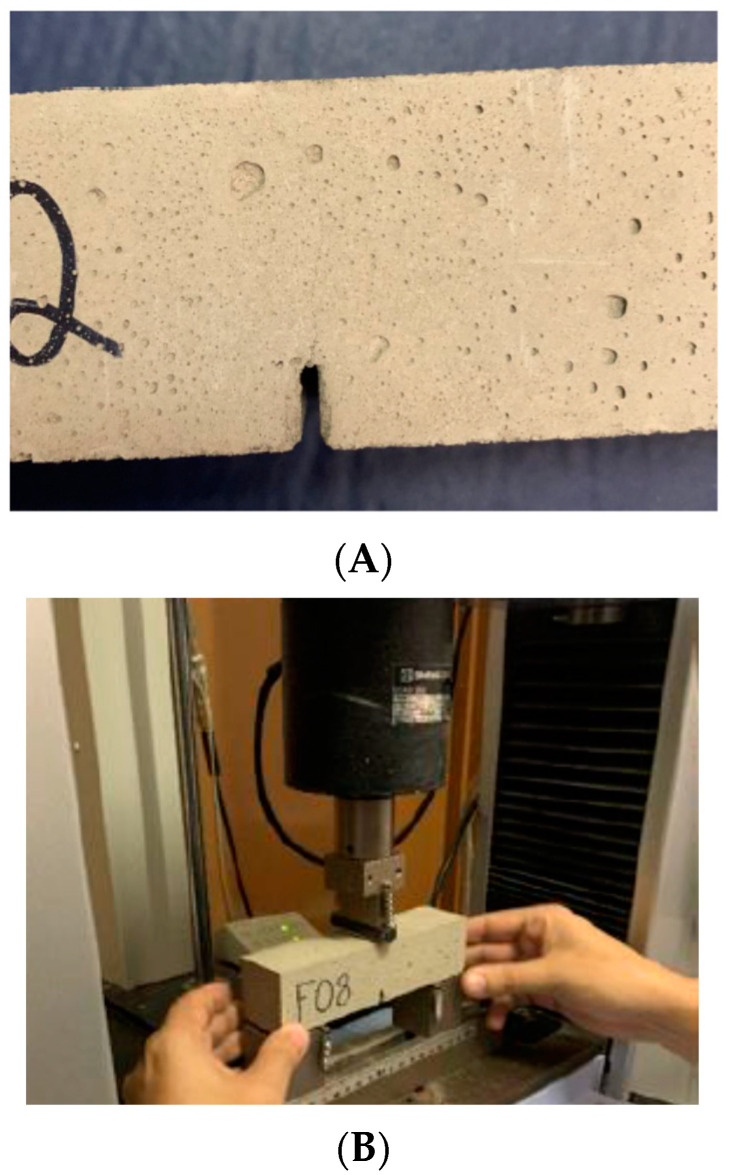
Prismatic sticky mortar samples: (**A**) detail of the slot with 9 mm height and (**B**) example of a sample fitted for the tests and the equipment used.

**Figure 2 materials-15-02332-f002:**
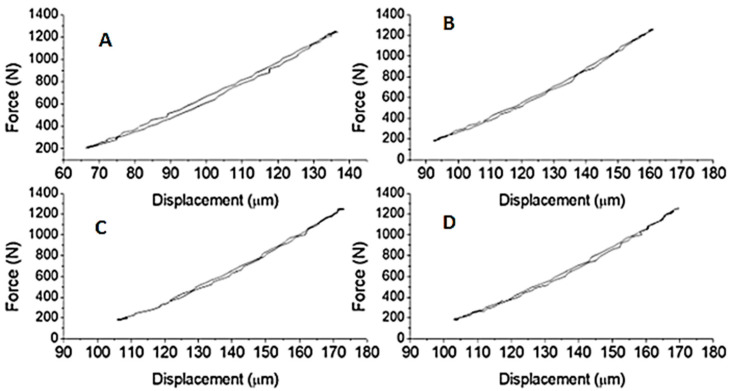
Hysteresis curves of the bending tensile strength test with cyclic loads of Solid-B samples: (**A**) 1000 cycles, (**B**) 2000 cycles, (**C**) 3000 cycles, and (**D**) 4000 cycles.

**Figure 3 materials-15-02332-f003:**
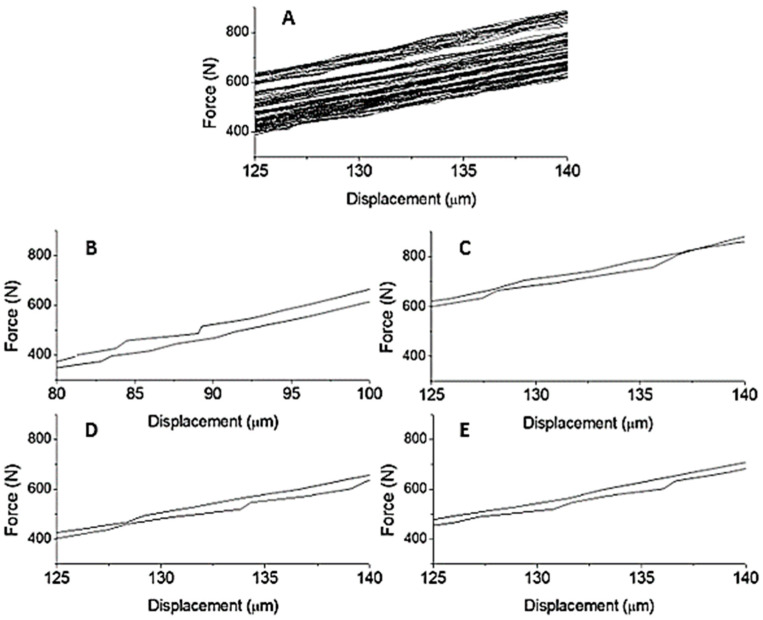
The central section of the hysteresis curves of the bending tensile strength test with cyclic loads of the Solid-B samples: (**A**) all 6000 cycles, (**B**) 1000 cycles, (**C**) 2000 cycles, (**D**) 3000 cycles, and (**E**) 4000 cycles.

**Figure 4 materials-15-02332-f004:**
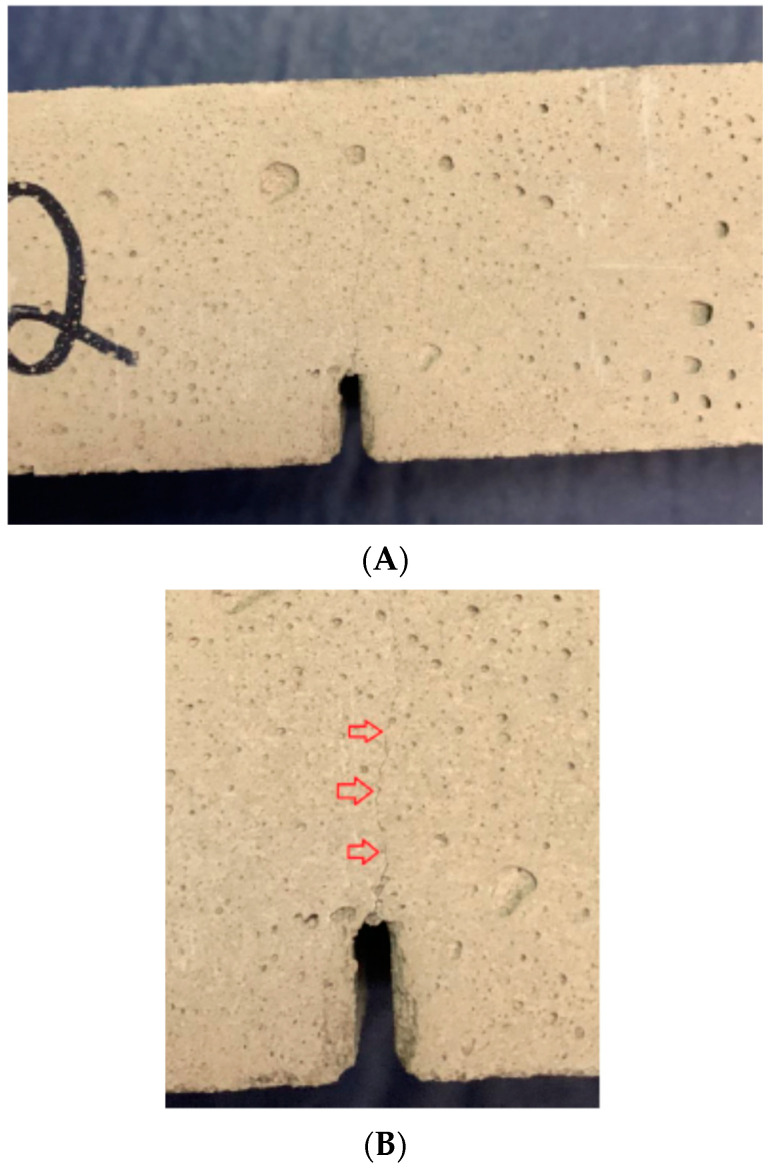
Single crack formed during the test: (**A**) sample after test; and (**B**) detail of the slot formed in the central axis of the previous failure.

**Table 1 materials-15-02332-t001:** Mechanical characterization tests of the adhesive mortar.

Tests	Experim. Results	Ref. Value (NBR 14081-4 [17])
Tensile adhesion strength, MPa (normal curing)	1.5	≥1.0
Tensile adhesion strength, MPa (immersed curing)	1.0	≥1.0
Tensile adhesion strength, MPa (steamed curing)	1.0	≥1.0
Open time, min	≥20	≥20

**Table 2 materials-15-02332-t002:** Experimental results of the bending tensile strength test.

Solid-B					Slot-B				
#	Max. Load (N)	Mean Load (N)	Max. Tension (MPa)	Mean Tension (MPa)	#	Max. Load (N)	Mean Load (N)	Max. Tension (MPa)	Mean Tension (MPa)
1	1705.9	1799.7	4.0	4.2	1	705.1	726.5	2.8	2.8
2	1775.3		4.2		2	729.3		2.8	
3	1899.8		4.4		3	760.4		3.0	
					4	763.8		3.0	
					5	712.8		2.8	
					6	687.6		2.7	

**Table 3 materials-15-02332-t003:** Average values of Solid-B and Slot-B mortar samples during the cyclic load test (values obtained from the central stretch of the hysteresis curve).

	Nr. Cycles	Load (N)		Unload (N)		Stress (MPa)		Energy (mJ)	
		F_min_	F_max_	F_min_	F_max_	Load	Unload	Elastic	Dissipated
Solid-B	1000	431.5	633.2	391.0	613.5	1.25	1.18	7.02	0.25
	2000	427.2	633.8	391.8	618.8	1.24	1.19	6.83	0.20
	3000	431.5	632.8	396.0	617.0	1.25	1.19	6.67	0.20
	4000	429.8	632.2	393.2	616.0	1.25	1.18	6.63	0.20
Slot-B	1000	176.8	257.4	161.7	250.1	0.85	0.80	1.55	0.03
	2000	176.7	257.7	160.0	250.8	0.85	0.80	1.55	0.03
	3000	176.2	258.6	161.4	251.0	0.85	0.80	1.55	0.04
	4000	175.3	256.8	160.9	249.6	0.84	0.80	1.51	0.03

**Table 4 materials-15-02332-t004:** Mechanical parameters.

Parameters	Solid-B	Slot-B	Decrease
I_c_ (m^4^)	2.13 × 10^−7^	0.99 × 10^−7^	54%
M_fc_ (kf.m)	2.17	0.89	60%
M_fd_ (kf.m)	2.07	0.85	59%
σ^c^ (MPa)	1.95	1.36	30%
σ^d^ (MPa)	1.90	1.29	32%

## Data Availability

The data that support the findings of this study are available upon request from the authors.

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
