# Peer review of "Mortar Bond Strength: A Brief Literature Review, Tests for Analysis, New Research Needs and Initial Experiments"

_materials, 2022, doi:10.3390/ma15062332_

Round 1
Reviewer 1 Report
It is an interesting study on the adhesive behavior of mortars. Before the publication I would suggest the following corrections:
In the second paragraph of the Introduction you describe the production rate of ceramics in Brazil. I would suggest to add the data that would serve an insight in more global situation.
2.1 Materials: Please correct the superscript in the first sentence (mm3).
I suppose “Table 3. Experimental results of the bending tensile strength test” on page 7 should be labeled as Table2? Please check the Table 3 subtitle.
Please unify the style of labels on all figures; (A) or (a), on figure or below the figure.
Please correct the label of Table 5.
Eq. (1) – please correct the position of (1).
Eq. (3) and (4) – Please use different label for the multiplication (dot is already a decimal separator and it is therefore unclear).
Author Response
Response to Reviewer 1 Comments
Dear Reviewer
We send one marked copy of the manuscript in which it may be seen that all the suggestions have been taken into account. The corrections are marked in the electronic version of the paper that was improved.
Reviewer#1
It is an interesting study on the adhesive behavior of mortars. Before the publication I would suggest the following corrections:
Point 1: In the second paragraph of the Introduction you describe the production rate of ceramics in Brazil. I would suggest to add the data that would serve an insight in more global situation.
Answer: Thanks for your suggestion. The following sentence was added: “the world total global production volume of ceramic tiles amounted to 16.09 thousand million square meters, in 2020”
Point 2: 2.1 Materials: Please correct the superscript in the first sentence (mm3).
Answer: Thanks for your suggestion. The sentence was corrected to “40x40x160 mm3”.
Point 3: I suppose “Table 3. Experimental results of the bending tensile strength test” on page 7 should be labeled as Table2? Please check the Table 3 subtitle.
Answer: Thanks for your suggestion. Old Table 3 was labeled as Table 2.
Point 4: Please unify the style of labels on all figures; (A) or (a), on figure or below the figure.
Answer: Thanks for your suggestion. All the figures were labeled uniformly.
Point 5: Please correct the label of Table 5.
Answer: Thanks for your suggestion. The label of Table 5 was corrected to “Table 5 -Values of the Spearman’s correlation coefficients.”
Point 6: Eq. (1) – please correct the position of (1).
Answer: Thanks for your suggestion. The position of (1) was corrected.
Point 7: Eq. (3) and (4) – Please use different label for the multiplication (dot is already a decimal separator and it is therefore unclear).
Answer: Thanks for your suggestion. The dot was replaced by “x” in Eqs. (3) and (4).
We trust that the manuscript will meet with your approval, but should any doubt remain, please let us know.
Thank you for your attention.
Best regards,
Janaina Salustio, Sandro M. Torres, Anne C. Melo, Ângelo J. Costa e Silva, António C. Azevedo, Jennef C. Tavares, Matheus S. Leal and João M.P.Q. Delgado
Reviewer 2 Report
apologies for typos and lack of capitals: my wrist is broken.
organization is a weakness: improvement is needed. sects 1.1 to 1.3 are background. the purpose was stated in the intro above these subsections. then in section 2 on methods, the purpose was restated, twice.
word choice and sentence structure are both problems. I’ve tried to point out parts I found most confusing.
details:
abst. this phrase is difficult to understand: “ the existing gaps related to the adhesion ability to adhesive mortars.”. to? or of?
this is confusing: “the values of stress, elastic, and dissipated energy were measured”
do they mean stress energy or stress? elastic energy or elasticity? Actually stress was measured and energy was calculated.
intro
l.1 what coatings? an intro must be independent of the abstract. this pp should be after or merged with the next two
end of first pp. the sentence is redundant.
3rd pp. it is not clear if the problem is coatings detachment or tile detachment. the abstract talks about coatings, but from this pp I think it is the tiles. tiles themselves have a coating or glaze.
1.1 “For Petit et al.” ??From Petit?
this sentence is a run-on.
1.2 I think this can be shortened. basically it says bonding is complex and has many aspects.
1.3 the 7th pp discussed repair mortors. seems off the topic.
- the first two pp belong in the intro.
2.1 in this section I realized that mortar is being tested. it is not failure of coatings that is being examined but the mortar itself. this needs to be clear earlier.
2.2a. is young’s modulus being measured?
table 1 does not align with the categories listed above.
pp below tab 1.: significant figure problem exists. for example, the first value should be 9.9+/-0.7 gpa. the rest of the digits are meaningless.
Section 2. it might be better to describe the tests, rather than discuss a long list of standard methods. If these are on the web, links would be useful.
3.1
The 2nd sentence is ackward.
Pp starting with figure 2: “Should be mentioned…” is not a sentence.
Next pp on energy analysis is poorly written. As it stands, I am not convinced energy was properly extracted.
“greater reflection zone” is not explained.
“materials even when requested below their elastic limit can…” is confusing. Tested? Instead of requested?
“These indications provide evidence that there is no true elastic limit” I am not sure this is shown convincingly in this pp. It seems like an aside.
3.3
“it is possible to consider that
the measurements obtained experimentally exceed the mechanical analysis performed.”
I find this sentence baffling.
The physical meaning of the equations is not clear.
With only 2 types studied, how can this be a variable in an equation? It is not surprising that different mortars behave differently. One might quantify behavior in terms of density or chemical composition.
4
The conclusion section is difficult to follow and seems overly detailed. The message is lost.
Author Response
Dear Reviewer
We send one marked copy of the manuscript in which it may be seen that all the suggestions have been taken into account. The corrections are marked in the electronic version of the paper that was improved.
Reviewer#2
Point 1: Organization is a weakness: improvement is needed. sects 1.1 to 1.3 are background. the purpose was stated in the intro above these subsections. then in section 2 on methods, the purpose was restated, twice.
Answer: Thanks for your suggestion. The paper organization was improved and objectives are not repeated across the paper. The objectives were presented only at the end of the introduction section.
Point 2: word choice and sentence structure are both problems. I’ve tried to point out parts I found most confusing. Abst. this phrase is difficult to understand: “ the existing gaps related to the adhesion ability to adhesive mortars.”. to? or of?
Answer: Thanks for your suggestion. The authors improve the paper, namely the sentence structure, in accordance with Reviewer suggestions. The sentence was improved to “the existing gaps related to the adhesion ability of adhesive mortars”.
Point 3: This is confusing: “the values of stress, elastic, and dissipated energy were measured”. Do they mean stress energy or stress? elastic energy or elasticity? Actually stress was measured and energy was calculated.
Answer: Thanks for your suggestion. The sentence was improved: “the values of stress, elastic energy, and dissipated energy were obtained”
Point 4: l.1 what coatings? an intro must be independent of the abstract. this pp should be after or merged with the next two
Answer: Thanks for your suggestion. The word “coatings” was replaced by “ceramic tiles” and the first 3 paragraphs were merged.
Point 5: end of first pp. the sentence is redundant.
Answer: Thanks for your suggestion. The sentence “These requirements are essential for the perfect functioning and meeting the minimum criteria of performance and functioning of the building” was deleted.
Point 6: 3rd pp. it is not clear if the problem is coatings detachment or tile detachment. the abstract talks about coatings, but from this pp I think it is the tiles. tiles themselves have a coating or glaze.
Answer: Thanks for your suggestion. The following sentence was added: “are several factors related to the tiles detachment”
Point 7: 1.1 “For Petit et al.” ??From Petit? This sentence is a run-on. I think this can be shortened. Basically it says bonding is complex and has many aspects.
Answer: Thanks for your suggestion. The sentence was changed to “Petit et al. [4] showed that…”. Moreover, this paragraph was shortened and improved considerably, namely, with the highlights of this work.
Point 8: 1.3 the 7th pp discussed repair mortars. seems off the topic.
Answer: Thanks for your suggestion. The paragraph was deleted.
Point 9: the first two pp belong in the intro.
Answer: Thanks for your suggestion. The two paragraphs have been re-written to meet the expectations of section 2.
Point 10: 2.1 in this section I realized that mortar is being tested. it is not failure of coatings that is being examined but the mortar itself. this needs to be clear earlier.
Answer: Thanks for your suggestion. The body text was improved in order to clarify the research significance and the objectives. A new paragraph was presented at the end of the Introduction section and at the beginning of Section 2. Finally, Section 2.1 begins with a brief description of the tests carried out on prismatic mortar bars.
Point 11: 2.2a. is young’s modulus being measured?
Answer: Thanks for your comment. Young’s modulus was not measured, but the dynamic modulus of elasticity was determined using ultrasonic wave propagation.
Point 12: table 1 does not align with the categories listed above.
Answer: Thanks for your suggestion. Section 2.2 was improved and organized in order to reply to the Reviewer’s comment.
Point 13: pp below tab 1.: significant figure problem exists. for example, the first value should be 9.9+/-0.7 gpa. the rest of the digits are meaningless.
Answer: Thanks for your suggestion. The sentence was corrected to “were 9.9±0.7 GPa and 9.4±0.6 GPa, respectively”.
Point 14: Section 2. it might be better to describe the tests, rather than discuss a long list of standard methods. If these are on the web, links would be useful.
Answer: Thanks for your suggestion. The body text was improved.
Point 15: Section 3.1 - The 2nd sentence is ackward.
Answer: Thanks for your suggestion. The sentence was improved to: “The experimental results of the tests with static loading showed that the highest values of maximum load are verified in the samples without the presence of the fault (ACIII-NC).”
Point 16: Pp starting with figure 2: “Should be mentioned…” is not a sentence.
Answer: Thanks for your suggestion. The sentence was improved to: “Only six samples of ACIII-YC and three samples of ACIII-NC reached the limit of 6000 cycles, the other ones ruptured before reaching it.”
Point 17: Next pp on energy analysis is poorly written. As it stands, I am not convinced energy was properly extracted.
Answer: Thanks for your suggestion. The paragraph was re-written in accordance with the Reviewer’s suggestion and added to the end of section 2.2.
Point 18: “greater reflection zone” is not explained.
Answer: Thanks for your suggestion. This sentence was deleted.
Point 19: “materials even when requested below their elastic limit can…” is confusing. Tested? Instead of requested?
Answer: Thanks for your suggestion. The sentence was changed to “materials even when tested below their”.
Point 20: “These indications provide evidence that there is no true elastic limit” I am not sure this is shown convincingly in this pp. It seems like an aside.
Answer: Thanks for your suggestion. This sentence was deleted.
Point 21: Section 3.3: “it is possible to consider that the measurements obtained experimentally exceed the mechanical analysis performed.” I find this sentence baffling.
Answer: Thanks for your suggestion. The sentence was improved.
Point 22: The physical meaning of the equations is not clear.
Answer: Thanks for your suggestion. The equations were explained in the body text in accordance with the Reviewer’s suggestion.
Point 23: With only 2 types studied, how can this be a variable in an equation? It is not surprising that different mortars behave differently. One might quantify behavior in terms of density or chemical composition.
Answer: Thanks for your suggestion. The nomenclature used (sample type) refers to the two samples tested, i.e, with and without a pre-slit. As it is a qualitative independent variable, it was necessary to transform it into a categorical variable, which can be used in the constructed model. In order to clarify that the variable (sample type) dealt with two groups of samples, adjustments were made to the text. Please see in section 2.3 the sentence “… as a function of the sample type (ACIII-NC and ACIII-YC)”; and in section 3.3 the paragraph: “Since the stress imposed on the material was dependent on the type of sample group that was being tested…” and “ Equation (2) shows that the change in the test of a fail less sample group for a failed sample group (sample type) produces..”.
Point 24: Section 4: The conclusion section is difficult to follow and seems overly detailed. The message is lost.
Answer: Thanks for your excellent comments. This section was re-written and improved.
We trust that the manuscript will meet with your approval, but should any doubt remain, please let us know.
Thank you for your attention.
Best regards,
Janaina Salustio, Sandro M. Torres, Anne C. Melo, Ângelo J. Costa e Silva, António C. Azevedo, Jennef C. Tavares, Matheus S. Leal and João M.P.Q. Delgado
Reviewer 3 Report
Dear Authors,
the revised paper introduces sounti of scientific interest.
I ask you to strengthen the analytical aspect described in "Materials and Methods" and for this I enclose some additional non-binding bibliography.The physical meaning of the equations is not clear.
I would also ask you to amend diagram (A) in Figure 2.
- Modano, M., Fabbrocino, F., Gesualdo, A., Matrone, G., Farina, I., Fraternali, F. On the forced vibration test by vibrodyne, COMPDYN 2015 - 5th ECCOMAS Thematic Conference on Computational Methods in Structural Dynamics and Earthquake Engineering, 209-217, ISSN 00002015.
- Bossio A., Fabbrocino F., Monetta T., Lignola G. P., Prota A., Manfredi G., Bellucci F., Corrosion effects on seismic capacity of reinforced concrete structures, Corrosion Review (IF: 2.528), February 2019, Vol. 37, Issue: 1, Pages 45–56, ISSN (Online) 2191-0316, ISSN (Print) 0334-6005, DOI: 10.1515/corrrev-2018-0044.
- Fabbrocino F., Farina I., Modano M., Loading noise effects on the system identification of composite structures by dynamic tests with vibrodyne, COMPOSITES. PART B, ENGINEERING (IF: 6.864), 2017, Vol. 115, Pag. 376-383, ISSN: 1359-8368, DOI: 10.1016/j.compositesb.2016.09.032.
This paper is interesting because dynamic stress measurements are presented for different mortar blocks and on similar blocks, but with a perforated gap before measurements.
Experimental results showed that samples with a pre-existing defect have 77% of the elastic energy of intact blocks. The authors concluded that energy can be associated with the ability of the material to hold its parts together.
The conclusions draw the research well.
Best regards
Author Response
Dear Reviewer
We send one marked copy of the manuscript in which it may be seen that all the suggestions have been taken into account. The corrections are marked in the electronic version of the paper that was improved.
Reviewer#3
The revised paper introduces sounds of scientific interest.
Point 1: I ask you to strengthen the analytical aspect described in "Materials and Methods" and for this I enclose some additional non-binding bibliography.
Modano, M., Fabbrocino, F., Gesualdo, A., Matrone, G., Farina, I., Fraternali, F. On the forced vibration test by vibrodyne, COMPDYN 2015 - 5th ECCOMAS Thematic Conference on Computational Methods in Structural Dynamics and Earthquake Engineering, 209-217, ISSN 00002015.
Bossio A., Fabbrocino F., Monetta T., Lignola G. P., Prota A., Manfredi G., Bellucci F., Corrosion effects on seismic capacity of reinforced concrete structures, Corrosion Review (IF: 2.528), February 2019, Vol. 37, Issue: 1, Pages 45–56, ISSN (Online) 2191-0316, ISSN (Print) 0334-6005, DOI: 10.1515/corrrev-2018-0044.
Fabbrocino F., Farina I., Modano M., Loading noise effects on the system identification of composite structures by dynamic tests with vibrodyne, COMPOSITES. PART B, ENGINEERING (IF: 6.864), 2017, Vol. 115, Pag. 376-383, ISSN: 1359-8368, DOI: 10.1016/j.compositesb.2016.09.032.
Answer: As the Reviewer suggested, the section “Materials and Methods” was improved with new analytical aspects described by Eqs. (1) to (4), and more references were added.
Point 2: The physical meaning of the equations is not clear.
Answer: The equations added were explained.
Point 3: I would also ask you to amend diagram (A) in Figure 2.
Answer: Figure 2 (A) has been removed
Point 4: This paper is interesting because dynamic stress measurements are presented for different mortar blocks and on similar blocks, but with a perforated gap before measurements. Experimental results showed that samples with a pre-existing defect have 77% of the elastic energy of intact blocks. The authors concluded that energy can be associated with the ability of the material to hold its parts together. The conclusions draw the research well.
Answer: Thanks for your kindness.
We trust that the manuscript will meet with your approval, but should any doubt remain, please let us know.
Thank you for your attention.
Best regards,
Janaina Salustio, Sandro M. Torres, Anne C. Melo, Ângelo J. Costa e Silva, António C. Azevedo, Jennef C. Tavares, Matheus S. Leal and João M.P.Q. Delgado
Round 2
Reviewer 2 Report
The ms is improved but problems remain in
- Word use
- Significant figures
- A meaningless statistical analysis… this is not based on any physical model and presumes sample type has a numerical value.
Below are suggestions re parts that need some work. Editing by a native English speaker may still be needed. I fixed what caught my attention. Essentially, he authors revision addressed my specific remarks, but did not go further to clean up similar errors.
Abstract
I think “ An experimental campaign with intact samples (without failure) and samples with a previous crack (failure) was presented and the values of stress energy, elastic energy, and dissipated energy were measured.”
should be
“Dynamic stress measurements are presented for several blocks of mortar and on similar blocks but with a crack drilled prior to measurements (intended to represent failure). From these data we calculated values of stress energy, elastic energy, and dissipated energy.”
This is confusing: “ The experimental results and the statistical analysis showed that the presence of failures reduced by 77% the elastic energy of the samples with the previous defect, showing that the energy can be associated with the material's ability to keep its parts together.”
Maybe “ The experimental results and the statistical analysis showed samples with a preexisting defect have 77% of the elastic energy of the intact blocks. We conclude that energy can be associated with the material's ability to keep its parts together.”
Intro, line 1. After tiles, add “on buildings”
“ Ahead Brazil, only China, with more than 5 thousand million square meters, and India, with just over 1.2 thousand million, are produced of more ceramic tiles.”
I think they mean
” Ahead of Brazil are China, which produced more than 5 thousand million square meters of ceramic tiles, and India, which produced just over 1.2 thousand million square meters.”
2.1 “pre-crack” should be “a slit drilled prior to measurements”
“pre-slit” should be ”This pre-existing slit”
“pre-dosed” ?? not sure what is meant.
I think theACIII-YC and ACIIi_NC names are confusing. It would be clearer to refer to these as Solid blocks and Notched blocks.
Notch or slot are probably better descriptors than slit. Crack is not appropriate.
I am guessing that ACIII has meaning, but was this explained?
Significant figure problems remain.
Below Table 1. “5.57 % should be 6%. Please check significant figures elsewhere, e.g, p-value.
Other Tables have digits that are not believable.
Below fig. 2, “requested” remains. This sentence needs work. Also, the proper term for failure due to cyclical stress is “fatigue” – see Schijve, J., 2009, Fatigue of Structures and Materials (2nd ed. with CD-ROM):
Dordrecht, Netherlands, Springer Science + Business Media, 621 p.,
ISBN 978-1-4020-6807-2.
Below fig 3 “pre-failure” ?? I think they mean “with a pre-existing slit”
3.2 My scientific concern is that the slit is fat. It is not a crack , but has corners. Cracks in the images differ. I think the authors should stick to slit as describing the drilling. Maybe slot or notch would be better, but not crack.
Section 3.3
Equations. Sample type remains used as a variable. This is arbitrary. Values assigned should be specified.
Table 5. The only strong correlation seems to be of loading and unloading stress. This may be an experimental constraint.
A moderate correlation exists between load and energy. This is bizarre –since: Energy = force x distance, force should be directly proportional to energy. The exponential for all variables reveals the flawed nature of this approach,
I doubt that the statistical analysis has any validity.
The results are in table 4. The statistical analysis adds nothing, and in fact detracts from the achievements of the paper.
The authors should remove this section, table 5, etc.
Author Response
Response to Reviewer 2 Comments
Dear Reviewer
We send one marked copy of the manuscript in which it may be seen that all the suggestions have been taken into account. The corrections are marked in the electronic version of the paper that was improved.
Reviewer#2
Point 1: The ms is improved but problems remain in Word use and Significant figures. A meaningless statistical analysis… this is not based on any physical model and presumes sample type has a numerical value.
Answer: Thanks for your excellent comments. Section 3.3 “statistical analysis was deleted in accordance with Reviewer suggestions.
Below are suggestions re parts that need some work. Editing by a native English speaker may still be needed. I fixed what caught my attention. Essentially, the author’s revision addressed my specific remarks but did not go further to clean up similar errors.
Point 2: Abstract: I think “An experimental campaign with intact samples (without failure) and samples with a previous crack (failure) was presented and the values of stress-energy, elastic energy, and dissipated energy were measured.” should be “Dynamic stress measurements are presented for several blocks of mortar and on similar blocks but with a crack drilled prior to measurements (intended to represent failure). From these data we calculated values of stress energy, elastic energy, and dissipated energy.”
Answer: Thanks for your comments. The sentence was improved in accordance with the Reviewer suggestion
Point 3: This is confusing: “The experimental results and the statistical analysis showed that the presence of failures reduced by 77% the elastic energy of the samples with the previous defect, showing that the energy can be associated with the material's ability to keep its parts together.” Maybe “The experimental results and the statistical analysis showed samples with a preexisting defect have 77% of the elastic energy of the intact blocks. We conclude that energy can be associated with the material's ability to keep its parts together.”
Answer: Thanks for your comments. The sentence was improved in accordance with the Reviewer suggestion
Point 4: Intro, line 1. After tiles, add “on buildings”
Answer: Thanks for your comments. The sentence was improved in accordance with the Reviewer suggestion
Point 5: “Ahead Brazil, only China, with more than 5 thousand million square meters, and India, with just over 1.2 thousand million, are produced of more ceramic tiles.” I think they mean “Ahead of Brazil are China, which produced more than 5 thousand million square meters of ceramic tiles, and India, which produced just over 1.2 thousand million square meters.”
Answer: Thanks for your comments. The sentence was improved in accordance with the Reviewer suggestion
Point 6: 2.1 “pre-crack” should be “a slit drilled prior to measurements”; “pre-slit” should be ”This pre-existing slit”; “pre-dosed” ?? not sure what is meant. I think the ACIII-YC and ACIIi_NC names are confusing. It would be clearer to refer to these as Solid blocks and Notched blocks. Notch or slot are probably better descriptors than a slit. Crack is not appropriate. I am guessing that ACIII has meaning, but was this explained?
Answer: Thanks for your comments. The expressions and sentences were improved in accordance with Reviewer suggestions and the term “pre-dosed” was deleted. Finally, the name “ACIII-YC” was changed to “Slot-B” and the name “ACIII-NC” was changed to “Solid-B”.
Point 7: Significant figure problems remain. Below Table 1. “5.57 % should be 6%. Please check significant figures elsewhere, e.g, p-value. Other Tables have digits that are not believable.
Answer: Thanks for your comments. Your suggestions were added.
Point 8: Below fig. 2, “requested” remains. This sentence needs work. Also, the proper term for failure due to cyclical stress is “fatigue” – see Schijve, J., 2009, Fatigue of Structures and Materials (2nd ed. with CD-ROM): Dordrecht, Netherlands, Springer Science + Business Media, 621 p., ISBN 978-1-4020-6807-2.
Answer: Thanks for your comments. The sentence was improved and the reference was added.
Point 9: Below fig 3 “pre-failure” ?? I think they mean “with a pre-existing slit”
Answer: Thanks for your comments. The sentence was improved in accordance with the Reviewer suggestion
Point 10: 3.2 My scientific concern is that the slit is fat. It is not a crack , but has corners. Cracks in the images differ. I think the authors should stick to slit as describing the drilling. Maybe slot or notch would be better, but not crack.
Answer: Thanks for your comments. The term “crack” was replaced by “slot”
Point 11: Section 3.3 - Equations. Sample type remains used as a variable. This is arbitrary. Values assigned should be specified. Table 5. The only strong correlation seems to be of loading and unloading stress. This may be an experimental constraint. A moderate correlation exists between load and energy. This is bizarre –since: Energy = force x distance, force should be directly proportional to energy. The exponential for all variables reveals the flawed nature of this approach, I doubt that the statistical analysis has any validity. The results are in table 4. The statistical analysis adds nothing, and in fact detracts from the achievements of the paper. The authors should remove this section, table 5, etc.
Answer: Thanks for your excellent comments. Section 3.3 “statistical analysis" was deleted in accordance with Reviewer suggestions.
We trust that the manuscript will meet with your approval, but should any doubt remain, please let us know.
Thank you for your attention.
Best regards,
Janaina Salustio, Sandro M. Torres, Anne C. Melo, Ângelo J. Costa e Silva, António C. Azevedo, Jennef C. Tavares, Matheus S. Leal and João M.P.Q. Delgado